# LEARNING MESH-BASED SIMULATION WITH GRAPH NETWORKS

**Tobias Pfaff**,[*] **Meire Fortunato**,[*] **Alvaro Sanchez-Gonzalez**,[*] **Peter W. Battaglia**
Deepmind, London, UK
{tpfaff,meirefortunato,alvarosg,peterbattaglia}@google.com

## ABSTRACT

Mesh-based simulations are central to modeling complex physical systems in many disciplines across science and engineering. Mesh representations support powerful numerical integration methods and their resolution can be adapted to strike favorable trade-offs between accuracy and efficiency. However, high-dimensional scientific simulations are very expensive to run, and solvers and parameters must often be tuned individually to each system studied. Here we introduce MESHGRAPHNETS, a framework for learning mesh-based simulations using graph neural networks. Our model can be trained to pass messages on a mesh graph and to adapt the mesh discretization during forward simulation. Our results show it can accurately predict the dynamics of a wide range of physical systems, including aerodynamics, structural mechanics, and cloth. The model's adaptivity supports learning resolution-independent dynamics and can scale to more complex state spaces at test time. Our method is also highly efficient, running 1-2 orders of magnitude faster than the simulation on which it is trained. Our approach broadens the range of problems on which neural network simulators can operate and promises to improve the efficiency of complex, scientific modeling tasks.

## 1 INTRODUCTION

State-of-the art modeling of complex physical systems, such as deforming surfaces and volumes, often employs mesh representations to solve the underlying partial differential equations (PDEs). Mesh-based finite element simulations underpin popular methods in structural mechanics [31, 48], aerodynamics [13, 34], electromagnetics [32], geophysics [35, 39], and acoustics [26]. Meshes also support adaptive representations, which enables optimal use of the resource budget by allocating greater resolution to regions of the simulation domain where strong gradients are expected or more accuracy is required, such as the tip of an airfoil in an aerodynamics simulation. Adaptive meshing enables running simulations at accuracy and resolution levels impossible with regular discretization schemes [8, 27] (Figure 3b).

Despite their advantages, mesh representations have received relatively little attention in machine learning. While meshes are sometimes used for learned geometry processing [9] and generative models of shapes [15, 29], most work on predicting high-dimensional physical systems focuses on grids, owing to the popularity and hardware support for CNN architectures [19]. We introduce a method for predicting dynamics of physical systems, which capitalizes on the advantages of adaptive mesh representations. Our method works by encoding the simulation state into a graph, and performing computations in two separate spaces: the mesh-space, spanned by the simulation mesh, and the Euclidean world-space in which the simulation manifold is embedded (see Figure 3a). By passing messages in mesh-space, we can approximate differential operators that underpin the internal dynamics of most physical systems. Message-passing in world-space can estimate external dynamics, not captured by the mesh-space interactions, such as contact and collision. Unstructured irregular meshes, as opposed to regular grids, support learning dynamics which are independent of resolution, allowing variable resolution and scale at runtime. By learning a map of desired resolution over the mesh (sizing field), together with a local remesher, our method can even adaptively change

---

[*]equal contribution

Videos of all our experiments can be found at https://sites.google.com/view/meshgraphnets

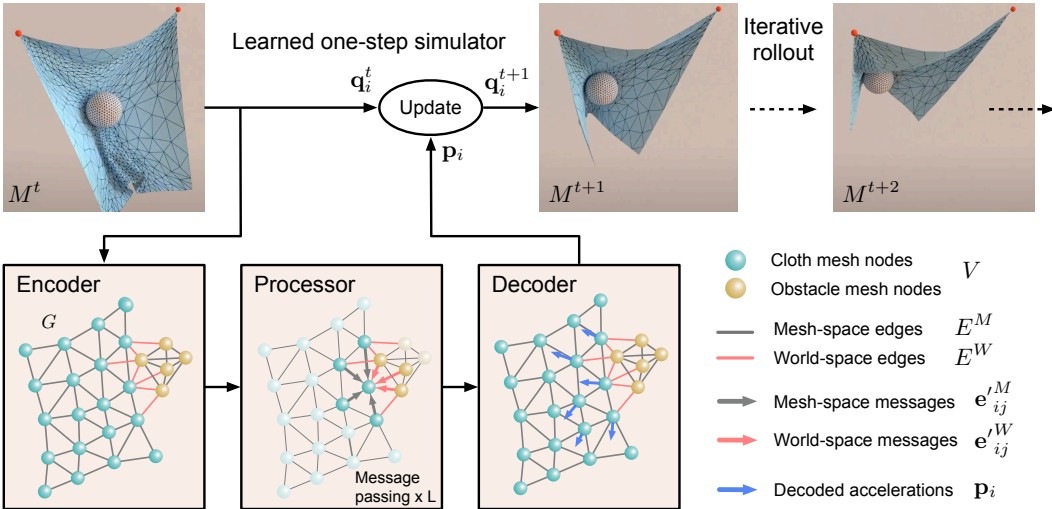

Figure 1: Diagram of MESHGRAPHNETS operating on our SPHEREDYNAMIC domain (video). The model uses an Encode-Process-Decode architecture trained with one-step supervision, and can be applied iteratively to generate long trajectories at inference time. The encoder transforms the input mesh $M^t$ into a graph, adding extra world-space edges. The processor performs several rounds of message passing along mesh edges and world edges, updating all node and edge embeddings. The decoder extracts the acceleration for each node, which is used to update the mesh to produce $M^{t+1}$.

the discretization during rollouts, budgeting greater computational resources for important regions of the simulation domain.

Together, our method allows us to learn the dynamics of vastly different physical systems, from cloth simulation over structural mechanics to fluid dynamics directly from data, providing only very general biases such as spatial equivariance. We demonstrate that by using mesh-space computation we can reliably model materials with a rest state such as elastics, which are challenging for mesh-free prediction models [37]. MESHGRAPHNETS outperform particle- and grid-based baselines, and can generalize to more complex dynamics than those on which it was trained.

## 2  RELATED WORK

Modelling high-dimensional physics problems with deep learning algorithms has become an area of great research interest in fields such as computational fluid dynamics. High resolution simulations are often very slow, and learned models can provide faster predictions, reducing turnaround time for workflows in engineering and science [16, 6, 49, 20, 1]. Short run times are also a desirable property for fluid simulation in visualization and graphics [46, 41, 47]. Learned simulations can be useful for real-world predictions where the physical model, parameters or boundary conditions are not fully known [12]. Conversely, the accuracy of predictions can be increased by including specialized knowledge about the system modelled in the form of loss terms [43, 23], or by physics-informed feature normalization [40].

The methods mentioned above are based on convolutional architectures on regular grids. Although this is by far the most widespread architecture for learning high-dimensional physical systems, recently there has been an increased interest in particle-based representations, which are particularly attractive for modelling the dynamics of free-surface liquids and granular materials. Ladicky et al. [22] use random forests to speed up liquid simulations. Various works [24, 42, 37] use graph neural networks (GNNs) [38, 4] to model particle-based granular materials and fluids, as well as glassy dynamics [3]. Learned methods can improve certain aspects of classical FEM simulations, e.g. more accurate handling of strongly nonlinear displacements [25] or learned elements which directly map between forces and displacements [10]. Finally, dynamics of high dimensional systems can be learned in reduced spaces. Holden et al. [18] performs PCA decomposition on cloth data, and

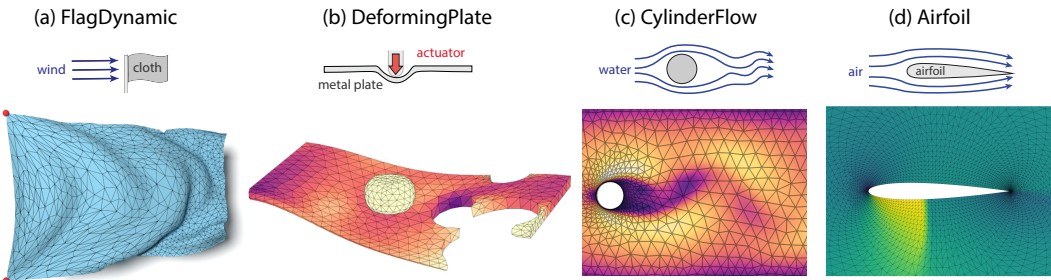

Figure 2: Our model can predict dynamics of vastly different physical systems, from structural mechanics over cloth to fluid dynamics. We demonstrate this by simulating (a) a flag waving in the wind, (b) a deforming plate, (c) flow of water around a cylinder obstacle, and (d) the dynamics of air around the cross-section of an aircraft wing (videos). The color map shows the von-Mises stress in (b), and the x-component of the velocity field in (c),(d).

learns a correction model to improve accuracy of subspace simulation. These models are however very domain-specific, and the expression range is limited due to the use of the linear subspace.

There is increased attention in using meshes for learned geometry and shape processing [9, 29, 17]. But despite mesh-based simulations being the tool of choice in mechanical engineering and related disciplines, adaptive mesh representations have not seen much use in machine learning for physics prediction, with a few notable exceptions [5, 2]. Belbute-Peres et al. [5] embed a differentiable aerodynamics solver in a graph convolution (GCN) [21] prediction pipeline for super-resolution in aerodynamics predictions. Our method has similarities, but without a solver in the loop, which potentially makes it easier to use and adapt to new systems. In Section 5 we show that MESH-GRAPHNETS are better suited for dynamical prediction than GCN-based architectures. Finally, Graph Element Networks [2] uses meshes over 2D grid domains to more efficiently compute predictions and scene representations. Notably they use small planar systems ($< 50$ nodes), while we show how to scale mesh-based predictions to complex 3D systems with thousands of nodes.

## 3 MODEL

We describe the state of the system at time $t$ using a simulation mesh $M^t = (V, E^M)$ with nodes $V$ connected by mesh edges $E^M$. Each node $i \in V$ is associated with a reference mesh-space coordinate $\mathbf{u}_i$ which spans the simulation mesh, and additional dynamical quantities $\mathbf{q}_i$ that we want to model. *Eulerian* systems (Figure 2c,d) model the evolution of continuous fields such as velocity over a fixed mesh, and $\mathbf{q}_i$ sample these fields at the mesh nodes. In *Lagrangian* systems, the mesh represents a moving and deforming surface or volume (e.g. Figure 2a,b), and contains an extra world-space coordinate $\mathbf{x}_i$ describing the dynamic state of the mesh in 3D space, in addition to the fixed mesh-space coordinate $\mathbf{u}_i$ (Figure 3a).

### 3.1 LEARNING FORWARD DYNAMICS

The task is to learn a forward model of the dynamic quantities of the mesh at time $t+1$ given the current mesh $M^t$ and (optionally) a history of previous meshes $\{M^{t-1}, ..., M^{t-h}\}$. We propose MESHGRAPHNETS, a graph neural network model with an Encode-Process-Decode architecture [4, 37], followed by an integrator. Figure 1 shows a visual scheme of the MESHGRAPHNETS architecture. Domain specific information on the encoding and integration can be found in Section 4.

**Encoder** The encoder encodes the current mesh $M^t$ into a multigraph $G = (V, E^M, E^W)$. Mesh nodes become graph nodes $V$, and mesh edges become bidirectional mesh-edges $E^M$ in the graph. This serves to compute the internal dynamics of the mesh. For Lagrangian systems, we add world edges $E^W$ to the graph, to enable learning external dynamics such as (self-) collision and contact,

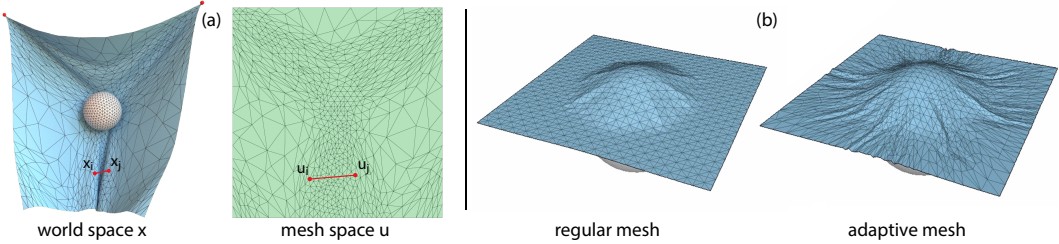

Figure 3: Simulation of a cloth interacting with a sphere. (a) In red, we highlight two nodes which are close in world-space but far in mesh-space, between which a world edge may be created. (b) With the same number of nodes, adaptive remeshing enables significantly more accurate simulations than a regular mesh with the same number of nodes.

which are non-local in mesh-space.[1] World-space edges are created by spatial proximity: that is, given a fixed-radius $r_W$ on the order of the smallest mesh edge lengths, we add a world edge between nodes $i$ and $j$ if $|\mathbf{x}_i - \mathbf{x}_j| < r_W$, excluding node pairs already connected in the mesh. This encourages using world edges to pass information between nodes that are spatially close, but distant in mesh space (Figure 3a).

Next, we encode features into graph nodes and edges. To achieve spatial equivariance, positional features are provided as relative edge features. We encode the relative displacement vector in mesh space $\mathbf{u}_{ij} = \mathbf{u}_i - \mathbf{u}_j$ and its norm $|\mathbf{u}_{ij}|$ into the mesh edges $\mathbf{e}_{ij}^M \in E^M$. Then, we encode the relative world-space displacement vector $\mathbf{x}_{ij}$ and its norm $|\mathbf{x}_{ij}|$ into both mesh edges $\mathbf{e}_{ij}^M \in E^M$ and world edges $\mathbf{e}_{ij}^W \in E^W$. All remaining dynamical features $\mathbf{q}_i$, as well as a one-hot vector indicating node type, are provided as node features in $\mathbf{v}_i$.

Finally, the concatenated features above are encoded into a latent vector of size 128 at each node and edge, using the encoder MLPs $\epsilon^M, \epsilon^W, \epsilon^V$ for mesh edges $\mathbf{e}_{ij}^M$, world edges $\mathbf{e}_{ij}^W$, and nodes $\mathbf{v}_i$ respectively. See sections 4 and A.1 for more details on input encoding.

**Processor** The processor consists of $L$ identical message passing blocks, which generalize Graph-Net blocks [36] to multiple edge sets. Each block contains a separate set of network parameters, and is applied in sequence to the output of the previous block, updating the mesh edge $\mathbf{e}_{ij}^M$, world edge $\mathbf{e}_{ij}^W$, and node $\mathbf{v}_i$ embeddings to $\mathbf{e}'^M_{ij}, \mathbf{e}'^W_{ij}, \mathbf{v}'_i$ respectively by

$$\mathbf{e}'^M_{ij} \leftarrow f^M(\mathbf{e}_{ij}^M, \mathbf{v}_i, \mathbf{v}_j), \quad \mathbf{e}'^W_{ij} \leftarrow f^W(\mathbf{e}_{ij}^W, \mathbf{v}_i, \mathbf{v}_j), \quad \mathbf{v}'_i \leftarrow f^V(\mathbf{v}_i, \sum_j \mathbf{e}'^M_{ij}, \sum_j \mathbf{e}'^W_{ij}) \quad (1)$$

where $f^M, f^W, f^V$ are implemented using MLPs with a residual connection.

**Decoder and state updater** For predicting the time $t+1$ state from the time $t$ input, the decoder uses an MLP $\delta^V$ to transform the latent node features $\mathbf{v}_i$ after the final processing step into one or more output features $\mathbf{p}_i$.

We can interpret the output features $\mathbf{p}_i$ as (higher-order) derivatives of $\mathbf{q}_i$, and integrate them using a forward-Euler integrator with $\Delta t = 1$ to compute the next-step dynamical quantity $\mathbf{q}_i^{t+1}$. For first-order systems the output $\mathbf{p}_i$ is integrated once to update $\mathbf{q}_i^{t+1} = \mathbf{p}_i + \mathbf{q}_i^t$, while for second-order integration happens twice: $\mathbf{q}_i^{t+1} = \mathbf{p}_i + 2\mathbf{q}_i^t - \mathbf{q}^{t-1}$. Additional output features $\mathbf{p}_i$ are also used to make direct predictions of auxiliary quantities such as pressure or stress. For domain-specific details on decoding, see Section 4. Finally, the output mesh nodes $V$ are updated using $\mathbf{q}_i^{t+1}$ to produce $M^{t+1}$. For some systems, we dynamically adapt the mesh after each prediction step; this is explained in the following section.

---

[1]From here on, any mention of world edges and world coordinates applies only to Lagrangian systems; they are omitted for Eulerian systems.

## 3.2 Adaptive Remeshing

Adaptive remeshing algorithms generally consist of two parts: identifying which regions of the simulation domain need coarse or fine resolution, and adapting the nodes and their connections to this target resolution. Only the first part requires domain knowledge of the type of physical system, which usually comes in the form of heuristics. For instance, in cloth simulation, one common heuristic is the refinement of areas with high curvature to ensure smooth bending dynamics (Figure 3b), while in computational fluid dynamics, it is common to refine around wall boundaries where high gradients of the velocity field are expected.

In this work we adopt the *sizing field* methodology [27]. The sizing field tensor $\mathbf{S}(\mathbf{u}) \in \mathbb{R}^{2 \times 2}$ specifies the desired local resolution by encoding the maximally allowed oriented, edge lengths in the simulation mesh. An edge $\mathbf{u}_{ij}$ is valid if and only if $\mathbf{u}_{ij}^{\mathrm{T}} \mathbf{S}_i \, \mathbf{u}_{ij} \leq 1$, otherwise it is too long, and needs to be split[2]. Given the sizing field, a generic local remeshing algorithm can simply split all invalid edges to refine the mesh, and collapse as many edges as possible, without creating new invalid edges, to coarsen the mesh. We denote this remeshing process as $M' = \mathcal{R}(M, \mathbf{S})$.

**Learned remeshing**  To leverage the advantages in efficiency and accuracy of dynamic remeshing, we need to be able to adapt the mesh at test time. Since remeshing requires domain knowledge, we would however need to call the specific remesher used to generate the training data at each step during the model rollout, reducing the benefits of learning the model. Instead, we learn a model of the sizing field (the only domain-specific part of remeshing) using the same architecture as in Section 3.1 and train a decoder output $\mathbf{p}_i$ to produce a sizing tensor for each node. At test time, for each time step we predict both the next simulation state and the sizing field, and use a generic, domain-independent remesher $\mathcal{R}$ to compute the adapted next-step mesh as $M^{t+1} = \mathcal{R}(\hat{M}^{t+1}, \hat{\mathbf{S}}^{t+1})$. We demonstrate this on triangular meshes, Section A.3 describes the simple generic remesher that we use for this purpose. While the sizing field is agnostic to the mesh type, other mesh types may require different local remeshers; for tetrahedral meshes a method such as Wicke et al. [45] could be used, while quad meshes can simply be split into triangular meshes.

## 3.3 Model Training

We trained our dynamics model by supervising on the per-node output features $\mathbf{p}_i$ produced by the decoder using a $L_2$ loss between $\mathbf{p}_i$ and the corresponding ground truth values $\bar{\mathbf{p}}_i$. Similarly, the sizing field model is trained with an $L_2$ loss on the ground truth sizing field. If sizing information is not available in the training data, e.g. not exposed by the ground truth simulator, we can still estimate a compatible sizing field from samples of simulator meshes, and use this estimate as labels (details in Section A.3.1).

## 4 Experimental Domains

We evaluated our method on a variety of systems with different underlying PDEs, including cloth, structural mechanics, incompressible and compressible fluids (Figure 2). Training and test data was produced by a different simulator for each domain. The simulation meshes range from regular to highly irregular: the edge lengths of dataset AIRFOIL range between $2 \cdot 10^{-4}$m to 3.5m, and we also simulate meshes which dynamically change resolution over the course of a trajectory. Full details on the datasets can be found in Section A.1.

Our structural mechanics experiments involve a hyper-elastic plate, deformed by a kinematic actuator, simulated with a quasi-static simulator (DEFORMINGPLATE). Both actuator and plate are part of the Lagrangian tetrahedral mesh, and are distinguished by a one-hot vector for the corresponding node type $\mathbf{n}_i$. We encode the node quantities $\mathbf{u}_i, \mathbf{x}_i, \mathbf{n}_i$ in the mesh, and predict the Lagrangian velocity $\dot{\mathbf{x}}_i$, which is integrated once to form the next position $\mathbf{x}_i^{t+1}$. As a second output, the model predicts the von-Mises stress $\sigma_i$ at each node.

Our cloth experiments involve a flag blowing in the wind (FLAGDYNAMIC) and a piece of cloth interacting with a kinematic sphere (SPHEREDYNAMIC) on an adaptive triangular mesh, which

---

[2]This formulation allows different maximal edge lengths depending on the direction. For e.g. a mesh bend around a cylinder, it allows to specify shorter edge lengths in the bent dimension than along the cylinder.

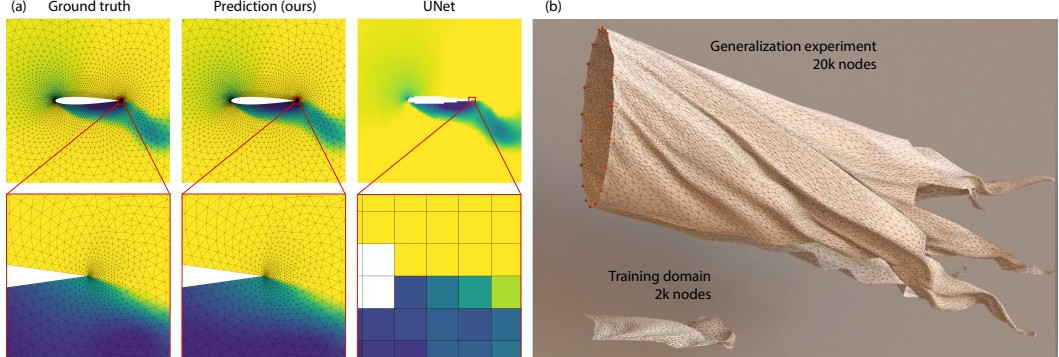

Figure 4: (a) Rollout of our model versus ground truth on dataset AIRFOIL. Adaptive meshing allows us to accurately predict dynamics at large and small scales. The grid-based U-Net baseline is capable of making good predictions at large scales, but it cannot resolve the smaller scales, despite using four times more cells than our model (video). (b) At inference time, our model can be scaled up to significantly larger and more complex setups than seen during training (video).

changes resolution at each time step. The dataset FLAGSIMPLE shares the setup of FLAGDY-NAMIC, but uses a static mesh and ignores collisions. The node type $\mathbf{n}_i$ distinguishes cloth and obstacle/boundary nodes, and we encode inputs $\mathbf{u}_i, \mathbf{x}_i, \mathbf{n}_i$ as above, but since this is a fully dynamic second order system, we additionally provide $h = 1$ steps of history, by including the velocity estimate $\dot{\mathbf{x}}_i^t = \mathbf{x}_i^t - \mathbf{x}_i^{t-1}$ as a node feature. The decoder outputs acceleration $\ddot{\mathbf{x}}_i$ which is integrated twice.

Our incompressible fluid experiments use the CYLINDERFLOW dataset, which simulates the flow of water around a cylinder on a fixed 2D Eulerian mesh. The mesh contains the node quantities $\mathbf{u}_i, \mathbf{n}_i, \mathbf{w}_i$, where $\mathbf{w}_i$ is a sample of the momentum field at the mesh nodes. In all fluid domains, the node type distinguishes fluid nodes, wall nodes and inflow/outflow boundary nodes. The network predicts change in momentum $\dot{\mathbf{w}}_i$, which is integrated once, and a direct prediction of the pressure field $p$.

Our compressible fluid experiments use the AIRFOIL dataset, which simulates the aerodynamics around the cross-section of an airfoil wing. We model the evolution of momentum[3] $\mathbf{w}$ and density $\rho$ fields, and hence the 2D Eulerian mesh encodes the quantities $\mathbf{u}_i, \mathbf{n}_i, \mathbf{w}_i, \rho_i$. We treat this as a first order system and predict change in momentum $\dot{\mathbf{w}}_i$ and density $\dot{\rho}_i$, as well as pressure $p_i$.

## 5 RESULTS

We tested our MESHGRAPHNETS model on our four experimental domains (Section 4), and compared it to three different baseline models. Our main findings are that MESHGRAPHNETS are able to produce high-quality rollouts on all domains, outperforming particle- and grid-based baselines, while being significantly faster than the ground truth simulator, and generalizing to much larger and more complex settings at test time.

Videos of rollouts, as well as comparisons, can be found at https://sites.google.com/view/meshgraphnets. Visually the dynamics remain plausible and faithful to the ground truth. Table 1 shows 1-step prediction and rollout errors in all of our datasets, while qualitative and quantitative comparisons are provided in Figure 4 and Figure 5. Even though our model was trained on next-step predictions, model rollouts remain stable for thousands of steps. This video shows a model trained on trajectories of 400 steps rolled out for 40000 steps.

**Learned remeshing** We trained both a dynamics and a sizing field model to perform learned dynamic remeshing during rollout on FLAGDYNAMIC and SPHEREDYNAMIC. We compare learned remeshing variants with sizing model learned from labeled sizing data, as in Section 3.2, as well as

---

[3]In visualizations, we show velocity, calculated as momentum $\mathbf{w}$ divided by density $\rho$.

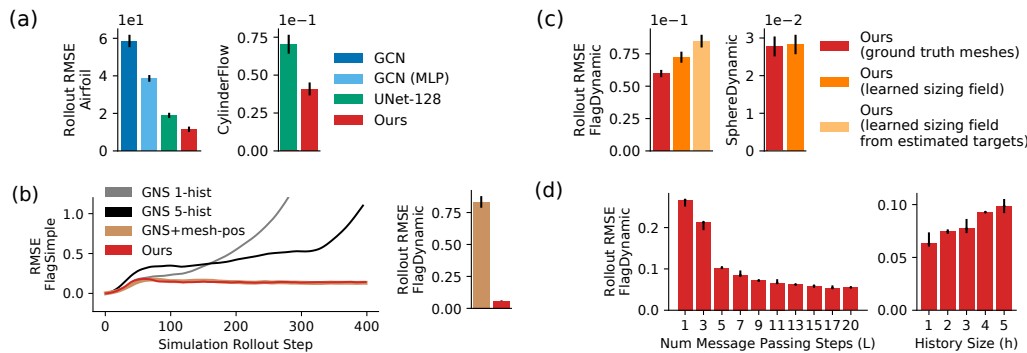

Figure 5: (a) Our model outperforms GCN and CNN-based baselines. (b) GNS diverges on cloth datasets; providing mesh-space positions (GNS+mesh-pos) helps, but still fails on dynamic meshes. (c) Remeshing with learned or estimated sizing fields produces accurate rollouts. (d) Taking sufficient message passing steps is crucial for good performance, and limiting history size increases accuracy by preventing overfitting.

from estimated targets, as in Section A.3.1. As a baseline, we ran our forward model on the ground truth mesh sequence. As observed in the video, all learned remeshing variants are able to shift the resolution to the new folds as they appear in the cloth, yield equally plausible dynamics, and are on par[4] in terms of quantitative performance (Figure 5c). Thus, our learned remeshing method provides the benefits of adaptive remeshing, which can be substantive in some domains, without requiring a domain-specific remesher in the loop.

**Computational efficiency** Our approach is consistently faster than ground truth solvers by one to two orders of magnitude on all domains (Table 1). We believe this is due to our model being able to take much larger timesteps than classical solvers, and avoiding performance bottlenecks. Additionally, classical general-purpose solvers on irregular domains, such as those studied in this paper, often do not scale well on hardware accelerators, while our model is built from neural network building blocks, highly suitable for hardware acceleration. A more detailed breakdown of performance on e.g. hardware setup is available in the appendix (section A.5.1). Our model's strong efficiency advantage means it may be applicable in situations where computing costs are otherwise prohibitive.

**Generalization** Our MESHGRAPHNETS model generalizes well outside of the training distribution, with respect to underlying system parameters, mesh shapes, and mesh size. This is because the architectural choice of using relative encoding on graphs has shown to be very conducive to

---

[4]Note that the comparison to ground truth requires interpolating to the ground truth mesh, incurring a small interpolation penalty for learned remeshing models.

| **Dataset** | **# nodes** (avg.) | **# steps** | $t_{model}$ ms/step | $t_{full}$ ms/step | $t_{GT}$ ms/step | **RMSE 1-step** $\times 10^{-3}$ | **RMSE rollout-50** $\times 10^{-3}$ | **RMSE rollout-all** $\times 10^{-3}$ |
|---|---|---|---|---|---|---|---|---|
| FLAGSIMPLE | 1579 | 400 | 19 | 19 | 4166 | $1.08 \pm 0.02$ | $92.6 \pm 5.0$ | $139.0 \pm 2.7$ |
| FLAGDYNAMIC | 2767 | 250 | 43 | 837 | 26199 | $1.57 \pm 0.02$ | $72.4 \pm 4.3$ | $151.1 \pm 5.3$ |
| SPHEREDYNAMIC | 1373 | 500 | 32 | 140 | 1610 | $0.292 \pm 0.005$ | $11.5 \pm 0.9$ | $28.3 \pm 2.6$ |
| DEFORMINGPLATE | 1271 | 400 | 24 | 33 | 2893 | $0.25 \pm 0.05$ | $1.8 \pm 0.5$ | $15.1 \pm 4.0$ |
| CYLINDERFLOW | 1885 | 600 | 21 | 23 | 820 | $2.34 \pm 0.12$ | $6.3 \pm 0.7$ | $40.88 \pm 7.2$ |
| AIRFOIL | 5233 | 600 | 37 | 38 | 11015 | $314 \pm 36$ | $582 \pm 37$ | $11529 \pm 1203$ |

Table 1: Left: Inference timings of our model per step on a single GPU, for pure neural network inference ($t_{model}$) and including remeshing and graph recomputation ($t_{full}$). Our model has a significantly lower running cost compared to the ground truth simulation ($t_{GT}$). A more detailed breakdown can be found in the section A.5.1. Right: Errors of our methods for a single prediction step (1-step), 50-step rollouts, and rollout of the whole trajectory.

generalization [37]. Also, by forcing the network to make predictions on very irregularly-shaped and dynamically changing meshes, we encourage learning resolution-independent physics.

In AIRFOIL, we evaluate the model on steeper angles ($-35°...35°$ vs $-25°...25°$ in training) and higher inflow speeds (Mach number $0.7...0.9$ vs $0.2...0.7$ in training). In both cases, the behavior remains plausible (video) and RMSE raises only slightly from $11.5$ at training to $12.4$ for steeper angles and $13.1$ for higher inflow speeds. We also trained a model on a FLAGDYNAMIC variant with wind speed and directions varying between trajectories, but constant within each trajectory. At inference time, we can then vary wind speed and direction freely (video). This shows that the local physical laws our models learns can extrapolate to untrained parameter ranges.

We also trained a model in the FLAGDYNAMIC domain containing only simple rectangular cloth, and tested its performance on three disconnected fish-shaped flags (video). Both the learned dynamics model and the learned remesher generalized to the new shape, and the predicted dynamics were visually similar to the ground truth sequence. In a more extreme version of this experiment, we test that same model on a windsock with tassels (Figure 4b, video). Not only has the model never seen a non-flat starting state during training, but the dimensions are also much larger — the mesh averages at 20k nodes, an order of magnitude more than seen in training. This result shows the strength of learning resolution and scale-independent models: we do not necessarily need to train on costly high-resolution simulation data; we may be able to learn to simulate large systems that would be too slow on conventional simulators, by training on smaller examples and scaling up at inference time. A more in-depth analysis on scaling can be found in the appendix A.5.3.

**Comparison to mesh-free GNS model**   We compared our method to the particle-based method GNS [37] on the fixed-mesh dataset FLAGSIMPLE to study the importance of mesh-space embedding and message-passing. As in GNS, the encoder builds a graph with fixed radius connectivity (10-20 neighbors per node), and relative world-space position embedded as edge features. As GNS lacks the notion of cloth's resting state, error accumulates dramatically and the simulation becomes unstable, with slight improvements if providing 5 steps of history (Figure 5b).

We also explored a hybrid method (GNS+mesh-pos) which adds a mesh-space relative position feature $\mathbf{u}_{ij}$ to the GNS edges. This yields rollout errors on par with our method (flattening after 50 steps due to decoherence in both cases), however, it tends to develop artifacts such as entangled triangles, which indicate a lack of reliable understanding of the mesh surface (video). On irregularly spaced meshes (FLAGSIMPLE), GNS+mesh-pos was not able to produce stable rollouts at all. A fixed connectivity radius will always oversample high-res regions, and undersample low-res regions of the mesh, leading to instabilities and high rollout errors (Figure 5b, right). We conclude that both having access to mesh-space positions as well as passing messages along the mesh edges are crucial for making predictions on irregularly spaced meshes.

Conversely, we found that passing message *purely* in mesh-space, without any world-space edges, also produces substandard results. On FLAGDYNAMIC and SPHEREDYNAMIC we observe an increase in rollout RMSE of $51\%$ and $92\%$ respectively, as (self-)collisions are harder to predict without world edges. In the latter case this is particularly easy to see: the obstacle mesh and cloth mesh are not connected, so without world edges, the model cannot compute their interaction at all.

**Comparison to GCNs**   To study the role of the graph network architecture, we tested our model against GCNs [21], which do not compute messages on edges. We adopted the GCN architecture from Belbute-Peres et al. [5] (without the super-resolution component) and trained it in the same setup as in our approach, including e.g. training noise and integration. We replicated results on the aerodynamical steady-state prediction task it was designed for (see Section A.4.2). On the much richer AIRFOIL task, however, GCN was unable to obtain stable rollouts. This is not simply a question of capacity; we created a hybrid (GCN-MLP) with our model (linear layers replaced by 2-hidden-layer MLPs + LayerNorm; 15 GCN blocks instead of 6), but the rollout quality was still poor (Figure 5a, video). We also ran an ablation of MESHGRAPHNETS without relative encoding in edges, for which absolute positional values are used as node features. This version performed much worse than our main model, yielding visual artifacts in the rollouts, and a rollout RMSE of $26.5$ in AIRFOIL. This is consistent with our hypothesis that the GCN performs worse due to the lack of relative encoding and message computing, which makes the GCN less likely to learn local physical laws and more prone to overfitting.

**Comparison to grid-based methods (CNNs)**   Arguably the most popular methods for predicting physical systems are grid-based convolutional architectures. It is fundamentally hard to simulate Lagrangian deforming meshes with such methods, but we can compare to grid-based methods on the Eulerian 2D domains CYLINDERFLOW and AIRFOIL, by interpolating the ROI onto a $128{\times}128$ grid. We implemented the UNet architecture from Thürey et al. [40], and found that on both datasets, MESHGRAPHNETS outperforms the UNet in terms of RMSE (Figure 5a). While the UNet was able to make reasonable predictions on larger scales on AIRFOIL, it undersampled the important wake region around the wingtip (Figure 4a), even while using four times more cells to span a region 16 times smaller than our method (Figure A.1). We observe similar behavior around the obstacle in CYLINDERFLOW. Additionally, as seen in the video, the UNet tends to develop fluctuations during rollout. This indicates that predictions over meshes presents advantages even in flat 2D domains.

**Key hyperparameters**   We tested several architecture variants and found our method is not very sensitive to many choices, such as latent vector width, number of MLP layers and their sizes. Nonetheless we identified two key parameters which influence performance (Figure 5d). Increasing the number of graph net blocks (message passing steps) generally improves performance, but it incurs a higher computational cost. We found that a value of 15 provides a good efficiency/accuracy trade-off for all the systems considered. Second, the model performs best given the shortest possible history (h=1 to estimate $\dot{x}$ in cloth experiments, h=0 otherwise), with any extra history leading to overfitting. This differs from GNS [37], which used $h \in 2...5$ for best performance.

## 6   CONCLUSION

MESHGRAPHNETS are a general-purpose mesh-based method which can accurately and efficiently model a wide range of physical systems, generalizes well, and can be scaled up at inference time. Our method may allow more efficient simulations than traditional simulators, and because it is differentiable, it may be useful for design optimization or optimal control tasks. Variants tailored to specific physical domains, with physics-based auxiliary loss terms, or energy-conserving integration schemes have the potential to increase the performance further. Finally, learning predictions on meshes opens the door for further work on resolution-adaptivity. For example, instead of learning adaptive meshing from ground truth data, we could learn a discretization which directly optimizes for prediction accuracy, or even performance on a downstream task. This work represents an important step forward in learnable simulation, and offers key advantages for modeling complex systems in science and engineering.

## ACKNOWLEDGMENTS

We would like to thank Danilo Rezende, Jonathan Godwin, Charlie Nash, Oriol Vinyals, Matt Hoffman, Kimberly Stachenfeld, Jessica Hamrick, Piotr Trochim, Emre Karagozler and our reviewers for valuable discussions, implementation help and feedback on the work and manuscript.

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

# A   APPENDIX

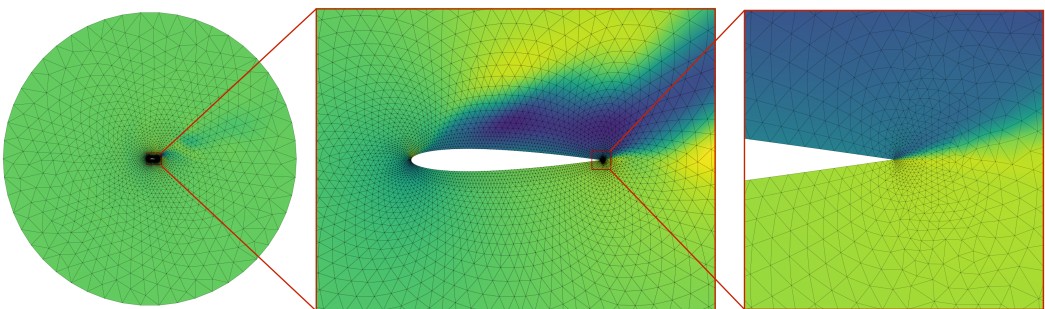

Figure A.1:   Many of our datasets have highly irregular meshing, which allows us to predict dynamics at several scales. With only 5k nodes, the dataset AIRFOIL spans a large region around the wing (left: entire simulation domain), while still providing high resolution around the airfoil (middle: ROI for visual comparison and RMSE computation), down to sub-millimeter details around the wing tip (right).

## A.1   DATASET DETAILS

Below we list details for all of our datasets. "System" describes the underlying PDE: cloth, hyperelasticity or compressible and incompressible Navier-Stokes flow. We used ArcSim [27] for simulating the cloth datasets, SU2 [13] for compressible flows, and COMSOL [11] for incompressible flow and hyperelastic simulations. Hyper-elasticity and cloth are simulated using linear elements. Each dataset consists of 1000 training, 100 validation and 100 test trajectories, each containing 250-600 time steps. Meshing can be either *regular*, i.e. all edges having similar length, *irregular*, i.e. edge lengths vary strongly in different regions of the mesh or *dynamic*, i.e. change at each step of the simulation trajectory. For Lagrangian systems, the world edge radius $r_W$ is provided. Our model operates on the simulation time step $\Delta t$ listed below. However, for each output time step, the solvers compute several internal time steps (16 for ArcSim, 100 for SU2, adaptive for COMSOL). As a quasi-static simulation, DEFORMINGPLATE does not have a time step.

| Dataset | System | Solver | Mesh type | Meshing | # steps | $\Delta$t s | $r_W$ |
|---|---|---|---|---|---|---|---|
| FLAGSIMPLE | cloth | ArcSim | triangle 3D | regular | 400 | 0.02 | — |
| FLAGDYNAMIC | cloth | ArcSim | triangle 3D | dynamic | 250 | 0.02 | 0.05 |
| SPHEREDYNAMIC | cloth | ArcSim | triangle 3D | dynamic | 500 | 0.01 | 0.05 |
| DEFORMINGPLATE | hyper-el. | COMSOL | tetrahedral 3D | irregular | 400 | — | 0.03 |
| CYLINDERFLOW | incompr. NS | COMSOL | triangle 2D | irregular | 600 | 0.01 | — |
| AIRFOIL | compr. NS | SU2 | triangle 2D | irregular | 600 | 0.008 | — |

Next, we list input encoding for mesh edges $\mathbf{e}_{ij}^M$, world edges $\mathbf{e}_{ij}^W$ and nodes $\mathbf{v}_i$, as well as the predicted output for each system.

| System | Type | inputs $\mathbf{e}_{ij}^M$ | inputs $\mathbf{e}_{ij}^W$ | inputs $\mathbf{v}_i$ | outputs $\mathbf{p}_i$ | history $h$ |
|---|---|---|---|---|---|---|
| Cloth | Lagrangian | $\mathbf{u}_{ij}, |\mathbf{u}_{ij}|, \mathbf{x}_{ij}, |\mathbf{x}_{ij}|$ | $\mathbf{x}_{ij}, |\mathbf{x}_{ij}|$ | $\mathbf{n}_i, (\mathbf{x}_i^t - \mathbf{x}_i^{t-1})$ | $\ddot{\mathbf{x}}_i$ | 1 |
| Hyper-El. | Lagrangian | $\mathbf{u}_{ij}, |\mathbf{u}_{ij}|, \mathbf{x}_{ij}, |\mathbf{x}_{ij}|$ | $\mathbf{x}_{ij}, |\mathbf{x}_{ij}|$ | $\mathbf{n}_i$ | $\dot{\mathbf{x}}_i, \sigma_i$ | 0 |
| Incomp. NS | Eulerian | $\mathbf{u}_{ij}, |\mathbf{u}_{ij}|$ | — | $\mathbf{n}_i, \mathbf{w}_i$ | $\dot{\mathbf{w}}_i, p_i$ | 0 |
| Compr. NS | Eulerian | $\mathbf{u}_{ij}, |\mathbf{u}_{ij}|$ | — | $\mathbf{n}_i, \mathbf{w}_i, \rho_i$ | $\dot{\mathbf{w}}_i, \dot{\rho}_i, p_i$ | 0 |

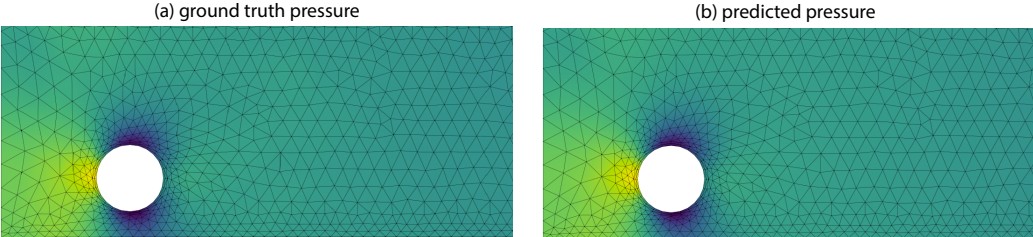

(a) ground truth pressure          (b) predicted pressure

Figure A.2: Beside output quantities such as position or momentum, which are integrated and fed back into the model as an input during rollout, we can also predict auxiliary output quantities, such as pressure or stress. These quantities can be useful for further analyzing the dynamics of the system. Here, we show a snapshot of auxiliary predictions of the pressure field in CYLINDERFLOW.

All second-derivative output quantities ($\ddot{\square}$) are integrated twice, while first derivative outputs ($\dot{\square}$) are integrated once as described in Section 3.1; all other outputs are direct predictions, and are not integrated. The one-hot node type vector $\mathbf{n}_i$ allows the model to distinguish between normal and kinematic nodes. Normal nodes are simulated, while kinematic either remain fixed in space (such as the two nodes which keep the cloth from falling), or follow scripted motion (as the actuator in DEFORMINGPLATE). For scripted kinematic nodes, we additionally provide the *next-step* world-space velocity $\mathbf{x}_i^{t+1} - \mathbf{x}_i^t$ as input; this allows the model to predict next-step positions which are consistent with the movement of the actuator. In the variant of FLAGDYNAMIC with varying wind speeds (generalization experiment in Section 5), the wind speed vector is appended to the node features.

In the dynamically meshed datasets (FLAGDYNAMIC, SPHEREDYANMIC), the mesh changes between steps, and there is no 1:1 correspondence between nodes. In this case, we interpolate dynamical quantities from previous meshes $M^{t-1}, ..., M^{t-h}$ as well as $M^{t+1}$ into the current mesh $M^t$ using barycentric interpolation in mesh-space, in order to provide history and targets for each node.

## A.2 ADDITIONAL MODEL DETAILS

### A.2.1 ARCHITECTURE AND TRAINING

The MLPs of the Encoder $\epsilon^M$, $\epsilon^W$, $\epsilon^V$, the Processor $f^M$, $f^W$, $f^V$, and Decoder $\delta^V$ are ReLU-activated two-hidden-layer MLPs with layer and output size of 128, except for $\delta^V$ whose output size matches the prediction $\mathbf{p}_i$. All MLPs outputs except $\delta^V$ are normalized by a LayerNorm. All input and target features are normalized to zero-mean, unit variance, using dataset statistics.

For training, we only supervise on the next step in sequence; to make our model robust to rollouts of hundreds of steps we use training noise (see Section A.2.2). Models are trained on a single v100 GPU with the Adam optimizer for 10M training steps, using an exponential learning rate decay from $10^{-4}$ to $10^{-6}$ over 5M steps.

### A.2.2 TRAINING NOISE

We used the same training noise strategy as in GNS [37] to make our model robust to rollouts of hundreds of steps. We add random normal noise of zero mean and fixed variance to the most recent value of the corresponding dynamical variable (Section A.2.3). When choosing how much noise to add, we looked at the one-step model error (usually related to the standard deviation of the targets in the dataset) and scanned the noise magnitude around that value on a logarithmic scale using two values for each factor of 10. For the exact numbers for each dataset, see Table A.2.3.

In the cases where the dataset is modelled as a first-order system (all, except cloth domains); we adjust the targets according to the noise, so that the model decoder produces an output that after integration would have corrected the noise at the inputs. For example, in DEFORMINGPLATE, assume the current position of a node is $x_i^t = 2$, and $\tilde{x}_i^t = 2.1$ after adding noise. If the next position is $x_i^{t+1} = 3$, the target velocity for the decoder $\dot{x}_i = 1$ will be adjusted to $\tilde{x}_i = 0.9$, so that after

integration, the model output $\tilde{x}_i^{t+1}$ matches the next step $x_i^{t+1}$ effectively correcting for the added noise, i.e. : $\tilde{x}_i^{t+1} = \tilde{x}_i^t + \tilde{\dot{x}}_i = 3 \equiv x_i^{t+1}$.

In the second-order domains (cloth), the model decoder outputs acceleration $\ddot{x}_i$ from the input position $x_i^t$ and velocity $\dot{x}_i^t = x_i^t - x_i^{t-1}$ (as in GNS). As with other systems, we add noise to the position $x_i^t$, which indirectly results on a noisy derivative $\dot{x}_i^t$ estimate. In this case, due to the strong dependency between position and velocity, it is impossible to adjust the targets to simultaneously correct for noise in both values. For instance, assume $x_i^{t-1} = 1.4, x_i^t = 2, x_i^{t+1} = 3$, which implies $\dot{x}_i^t = 0.6, \dot{x}_i^{t+1} = 1$, and ground truth acceleration $\ddot{x}_i = 0.4$. After adding 0.1 of noise the inputs are $\tilde{x}_i^t = 2.1 \Rightarrow \tilde{\dot{x}}_i^t = 0.7$. At this point, we could use a modified acceleration target of $\tilde{\ddot{x}}_i^P = 0.2$, so that after integration, the next velocity is $\tilde{\dot{x}}_i^{t+1} = \tilde{\dot{x}}_i^t + \tilde{\ddot{x}}^P = 0.9$, and the next position $\tilde{x}_i^{t+1} = \tilde{x}_i^t + \tilde{\dot{x}}_i^{t+1} = 3 \equiv x_i^{t+1}$, effectively correcting for the noise added to the position. However, note that in this case the predicted next step velocity $\tilde{\dot{x}}_i^{t+1} = 0.9$ does not match the ground truth $\dot{x}_i^{t+1} = 1$. Similarly, if we chose a modified target acceleration of $\tilde{\ddot{x}}_i^V = 0.3$, the next step velocity $\tilde{\dot{x}}_i^{t+1} = 1$ would match the ground truth, correcting the noise in velocity, but the same would not be true for the next step position $\tilde{x}_i^{t+1} = 3.1$. Empirically, we treated how to correct the noise for cloth simulation as a hyperparameter $\gamma \in [0, 1]$ which parametrizes a weighted average between the two options: $\tilde{\ddot{x}}_i = \gamma \tilde{\ddot{x}}_i^P + (1 - \gamma)\tilde{\ddot{x}}_i^V$. Best performance was achieved with $\gamma = 0.1$.

Finally, when the model takes more than one step of history ($h > 1$) (e.g. in the ablation from Figure 5d on FLAGDYNAMIC), the noise is added in a random walk manner with a per-step variance such as the variance at the last step matches the target variance (in accordance with GNS [37]).

### A.2.3 HYPERPARAMETERS

| Dataset | Batch size | Noise scale |
|---------|:----------:|:-----------:|
| FLAGSIMPLE | 1 | pos: 1e-3 |
| FLAGDYNAMIC | 1 | pos: 3e-3 |
| SPHEREDYNAMIC | 1 | pos: 1e-3 |
| DEFORMINGPLATE | 2 | pos: 3e-3 |
| CYLINDERFLOW | 2 | momentum: 2e-2 |
| AIRFOIL | 2 | momentum: 1e1, density: 1e-2 |

Table 2: Training noise parameters and batch size.

### A.3 A DOMAIN-INVARIANT LOCAL REMESHER FOR TRIANGULAR MESHES

A local remesher [27, 28, 33] changes the mesh by iteratively applying one of three fundamental operations: *splitting* an edge to refine the mesh, *collapsing* an edge to coarsen it, and *flipping* an edge to change orientation and to preserve a sensible aspect ratio of its elements. Edge splits create a new node whose attributes (position, etc.), as well as the associated sizing tensor, are obtained by averaging values of the two nodes forming the split edge. Collapsing removes a node from the mesh, while edge flips leave nodes unaffected.

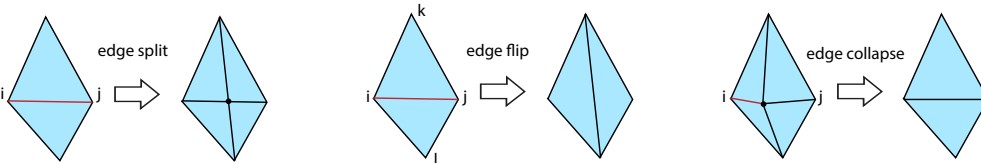

Given the sizing field tensor $\mathbf{S}_i$ at each node $i$, we can define the following conditions for performing edge operations:

- An edge connecting node $i$ and $j$ should be *split* if it is invalid, i.e. $\mathbf{u}_{ij}^T \mathbf{S}_{ij} \mathbf{u}_{ij} > 1$ with the averaged sizing tensor $\mathbf{S}_{ij} = \frac{1}{2}(\mathbf{S}_i + \mathbf{S}_j)$.

- An edge should be *collapsed*, if the collapsing operation does not create any new invalid edges.

- An edge should be *flipped* if the an-isotropic Delaunay criterion [7]

$$(\mathbf{u}_{jk} \times \mathbf{u}_{ik})\mathbf{u}_{il}^T \mathbf{S}_A \mathbf{u}_{jl} < \mathbf{u}_{jk}^T \mathbf{S}_A \mathbf{u}_{ik}(\mathbf{u}_{il} \times \mathbf{u}_{jl}), \qquad \mathbf{S}_A = \frac{1}{4}(\mathbf{S}_i + \mathbf{S}_j + \mathbf{S}_k + \mathbf{S}_l)$$

  is satisfied. This optimizes the directional aspect ratio of the mesh elements.

We can now implement a simple local remesher by applying these operations in sequence. First, we split all possible mesh edges to refine the mesh (in descending order of the metric $\mathbf{u}_{ij}^T \mathbf{S}_{ij} \mathbf{u}_{ij}$), then flip all edges which should be flipped. Next, we collapse all edges we can collapse (in ascending order of the metric $\mathbf{u}_{ij}^T \mathbf{S}_{ij} \mathbf{u}_{ij}$) to coarsen the mesh as much as possible, and finally again flip all possible edges to improve mesh quality.

### A.3.1 Estimating Sizing Field Targets

If no sizing field is available to train the sizing model, we can estimate it from a sequence of meshes. That is, for two consecutive meshes $M^t$, $M^{t+1}$ we want to find the sizing field $\mathbf{S}$ that would have induced this transition with a local remesher, i.e. $M^{t+1} = \mathcal{R}(M(t), \mathbf{S})$. To do this, we assume that the remesher is near-optimal, that is, all resulting edges are valid, yet maximum-length under the metric $\mathbf{S}$. For each $\mathbf{S}_i$ associated with the node $i$, this can be expressed as:

$$\mathbf{S}_i = \arg\max \sum_{j \in \mathcal{N}_i} \mathbf{u}_{ij}^T \mathbf{S}_i \, \mathbf{u}_{ij}, \quad s.t. \, \forall j \in \mathcal{N}_i : \mathbf{u}_{ij}^T \mathbf{S}_i \mathbf{u}_{ij} \leq 1 \tag{2}$$

This problem corresponds to finding the minimum-area, zero-centred ellipse containing the points $\mathbf{u}_{ij}$, and can be solved efficiently using the MINIDISK algorithm [44].

## A.4 Additional Baseline Details

### A.4.1 Baseline Training

Baseline architectures were trained within our general training framework, sharing the same normalization, noise and state-update strategies. We optimized the training hyperparameters separately in each case.

### A.4.2 GCN Baseline

We re-implemented the base GCN architecture (without the super-resolution component) from Belbute-Peres et al. [5]. To replicate the results, and ensure correctness of our implementation of the baseline, we created a dataset AIRFOILSTEADY which matches the dataset studied in their work. It uses the same solver and a similar setup as our dataset AIRFOIL, except that it has a narrower range of angle of attack ($-10°...10°$ vs $-25°...25°$ in AIRFOIL). The biggest difference is that the prediction task studied in their paper is not a dynamical simulation as our experiments, but a steady-state prediction task. That is, instead of unrolling a dynamics model for hundreds of time steps, this task consists of directly predicting the final steady-state momentum, density and pressure fields, given only two scalars (Mach number $m$, angle of attack $\alpha$) as well as the target mesh positions $\mathbf{u}_i$— essentially learning a parametrized distribution.

In AIRFOILSTEADY, the GCN predictions are visually indistinguishable to the ground truth, and qualitatively match the results reported in Belbute-Peres et al. [5] for their "interpolation regime" experiments. We also trained our model in AIRFOILSTEADY, as a one-step direct prediction model (without an integrator), with encoding like in AIRFOIL (see Section A.1), but where each node is conditioned on the global Mach number $m$ and angle of attack $\alpha$), instead of density and momentum. Again, results are visually indistinguishable from the ground truth (video), and our model outperforms GCN in terms of RMSE (ours 0.116 vs GCN 0.159). This is remarkable, as our models' spatial equivariance bias works against this task of directly predicting a global field. This speaks of the flexibility of our architecture, and indicates that it can be used for tasks beyond learning local physical laws for which it was designed.

### A.4.3 GRID (CNN) BASELINE

We re-implemented the UNet architecture of Thurey et al. [40] to exactly match their open-sourced version of the code. We used a batch size of 10. The noise parameters from Section A.2.3 are absolute noise scale on momentum 6e-2 for CYLINDERFLOW, and 1e1 on momentum and 1.5e-2 on density in the AIRFOIL dataset.

## A.5 ADDITIONAL ANALYSIS

### A.5.1 PERFORMANCE

In the table below, we show a detailed breakdown of per-step timings of our model run on CPU (8-core workstation) or a single v100 GPU. $t_{model}$ measures inference time of the graph neural network, while $t_{full}$ measures the complete rollout, including remeshing and graph recomputation. The ground truth simulation ($t_{GT}$) was run on the same 8-core workstation CPU. On our datasets, inference uses between 1-2.5GB of memory, including model variables and system overhead.

| Dataset | CPU $t_{model}$ ms/step | CPU $t_{full}$ ms/step | GPU $t_{model}$ ms/step | GPU $t_{full}$ ms/step | $t_{GT}$ ms/step | CPU speedup | GPU speedup |
|---|---|---|---|---|---|---|---|
| FLAGSIMPLE | 186 | 187 | 19 | 19 | 4166 | 22.3 | 214.7 |
| FLAGDYNAMIC | 534 | 1593 | 43 | 837 | 26199 | 16.4 | 31.3 |
| SPHEREDYNAMIC | 221 | 402 | 32 | 140 | 1610 | 4.0 | 11.5 |
| DEFORMINGPLATE | 172 | 174 | 24 | 33 | 2893 | 16.6 | 89.0 |
| CYLINDERFLOW | 166 | 168 | 21 | 23 | 820 | 4.9 | 35.3 |
| AIRFOIL | 497 | 499 | 37 | 38 | 11015 | 22.1 | 289.1 |

The NN bulding blocks used in our model are highly optimized for hardware acceleration. However, our ground truth solvers (ArcSim, COMSOL and SU2) do not support GPUs; and more broadly, solvers have varying levels of optimization for different hardware, so we find it hard to provide a 'true' hardware-agnostic performance comparison. We do note a few trends.

In the simulation regime studied in this paper (i.e. general-purpose simulations on complex, irregular domains) classical GPU solvers tend to be comparably hard to implement and they do not scale very well, thus many packages do not provide such support. As an example of a general-purpose solver with partial GPU support, ANSYS shows limited speedups of 2x-4x on GPU, even under optimal conditions [30, 14]. On the other hand, evaluating our model on the same CPU hardware as the ground truth solvers, it still achieves speedups between 4x-22x, even in this setting which is sub-optimal for NN models.

In practice, using a single GPU, we see speedups of 11x-290x compared to ArcSim, COMSOL and SU2, and users of such simulators with access to a GPU could benefit from these speedups.

### A.5.2 ERROR METRICS

Rollout RMSE is calculated as the root mean squared error of the position in the Lagrangian systems and of the momentum in the Eulerian systems, taking the mean for all spatial coordinates, all mesh nodes, all steps in each trajectory, and all 100 trajectories in the test dataset. The error bounds in Table 1 and the error bars in Figure 5(a-c) indicate standard error of the RMSE across 100 trajectories. Error bars in Figure 5(d) correspond to min/median/max performance across 3 seeds.

In FLAGSIMPLE and FLAGDYNAMIC, we observed decoherence after the first 50 steps (Figure 5b), due to the chaotic nature of cloth simulation. Since the dynamics of these domains are stationary, we use the rollout error in the first 50 steps of the trajectory for the comparison shown in the bar plots, as a more discerning metric for result quality. However, the reported trends also hold when measured over the whole trajectory.

In AIRFOIL, we compute the RMSE in a region of interest around the wing (Figure A.1 middle), which corresponds to the region shown in figures and videos. For comparisons with grid-based methods, we map the predictions on the grid to the ground truth mesh to compute the error.

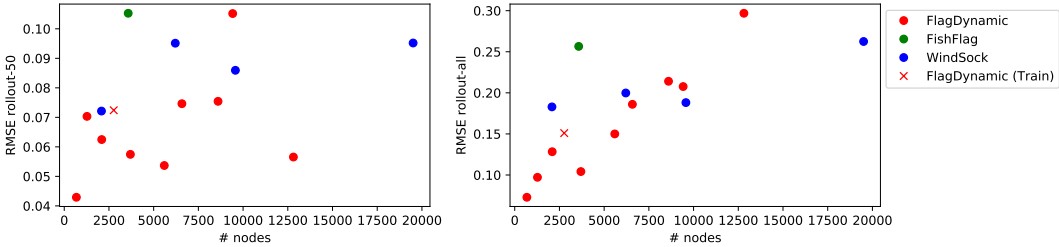

Figure A.3: A model trained on the regular-sized FLAGDYNAMIC domain was run on variants of FLAGDYNAMIC, WINDSOCK, FISHFLAG with different scale and resolutions. We show the RMSE for 50-step (left) and full-trajectory rollout (right) as a function of the simulation node count.

### A.5.3 ADDITIONAL ANALYSIS ON GENERALIZATION AND SCALING

We ran inference of our model trained on the FLAGDYNAMIC domain (with learned remeshing), on several scaled-up and scaled-down versions of FLAGDYNAMIC, and the generalization experiment WINDSOCK and FISHFLAG (see Section 5). In Figure A.3, we report the error compared to the respective ground-truth simulations.

When evaluating the 50-step RMSE rollout error in FLAGDYNAMIC we do not observe systematic trends of the error as function of the simulation size, indicating that the model performs similarly well on larger and smaller systems. The error when generalizing to new shapes (WINDSOCK, FISH-FLAG) is slightly higher, but comparable.

The RMSE rollout error evaluated on the full trajectory shows a stronger correlation with the system size. However, we believe this simply tracks the systematic positional error incurred due to decoherence (e.g. a small angle perturbation due to decoherence incurs a higher positional error at the tip of the flag the larger the flag is), and as shown in Figure 5b, decoherence becomes the main source of error after the first 50 steps of the simulation in this domain.

