# OpenReview forum: "Learning Mesh-Based Simulation with Graph Networks"
_ICLR.cc/2021/Conference — ICLR 2021 Spotlight_

### Official Review · AnonReviewer3 · 2020-10-26
**Possibly a game-changing paper for ML for scientific computing**

**Rating:** 10
**Confidence:** 4

**Review:**

The paper proposes a generic strategy to accelerate time integration of physical systems, including elasticity, fluid simulation, and cloth simulation. The results are impressive not only for  the generality, but because this is the first paper that I read that shows a fair speedup combined with stability for such complex physical systems. I wish the authors attached the source code so that I could analyze it more closely to make sure that no shortcuts were taken in the evaluation.

Assuming that the evaluation is correct, this is a seminal paper that will allow to use data-driven methods in scientific computing applications.

Comments:
- the entire discussion about adaptive refinement seems out of place. Would the method also work without and is this an extra feature? Or is it necessary to use this feature to make it work? My understanding is that this is optional, and only helps, in particular for cloth. Can you confirm that this is the case?
- I do not understand the sentence: “Each output feature is then processed by a forward-Euler integrator with ∆t = 1”. It only becomes clear after reading section 4. Please clarify earlier
- Why is the model stable for hundreds of steps? Is this due to training with noise? Please do experiment to try to identify which feature makes it stable. Will it eventually break if you let the system run for thousands of frames
- while the learning baseline are clearly described, I do not understand how the timings are measured for the GT in table 1. Is every row computed with a different software, as listed in A.1? If so, are all of these using a single core or multiple? This should be clarified, especially since the machine learning results are run on a gpu, making the comparison not fair. Could you report also the timing for doing the prediction of the NN using a single CPU core, or alternative run the iterative solvers of the classical methods on the same GPU?
- Is there a case where you could notice that the integration is not stable? Maybe by reducing the noise in the training or by reducing the training data size or by simulating scenes far from the training data? It would be interesting to know what the limitations of the method are and where there is room for improvement.
- Could you add a simple nearest neighbor baseline?
- It would be interesting to add a comparison of running time *for a given accuracy* instead of simply showing what is the error in the table. This could be achieved running many simulations with analytic methods until they match the error of the data-driven model wrt a finely discretized ground truth.

Overall, my impression is that the results look amazingly good: it is the first paper I read that shows that data-driven method can beat standard scicomp approaches on complex scenes. The reason for the performance of  the methods likely lies in the combination of using mesh and space variables, plus making the integration step explicit and enriching the training with controlled noise. I wish the authors attached the source code of  their method and of baselines (and I encourage them to do it after publication), I look forward to experimenting with this method and test it on my applications.

Update:
The revision further improved the paper and addressed most of my comments. I am still positive and voting for accepting this paper.

---

> ### Author Response · Authors · 2020-11-14
> **Response to R3**
>
> Thank you for your comments, questions and suggestions to improve the manuscript.
>
>
> - You are correct, adaptive remeshing at every step is optional, and our method works without it. Doing so however can significantly improve the simulation quality for domains like cloth, where the optimal discretization will change a lot during the rollout.
>
> - We will clarify the description of the integrator as suggested.
>
> - We believe the two key ingredients for long, stable rollouts are providing mesh-space coordinates, and training noise. Without mesh-space coordinates, the model may lose track of the system’s rest state, and rollouts will become unstable (fig. 5b, and GNS comparison video). And training noise is essential to make the model robust to build-up of small prediction errors; without training noise, all of our models will become unstable. Our models can remain stable for longer than the 100s of steps currently reported in the paper; we have tested this by rolling out a model for 40000 steps, and will include these results in the manuscript.
>
> -  t_gt in table 1 is the execution time of the ground truth solvers (Arcsim, COMSOL, SU2), which we ran on a typical 8-core workstation CPU. The solvers use multiple CPU cores, but do not support GPU acceleration-- this is quite common for general-purpose simulators for the type of domains we study. One of the advantages of our method is precisely that they allow us to strongly benefit from GPU/TPU accelerators, while most classical general-purpose simulators do not. We acknowledge that the claims about timing and speedups need further clarification, and we will include a more detailed report in the paper.
>
> - Besides training noise (as mentioned above), a minimum amount of diversity in the training set is necessary for stable generalization, more so than the absolute number of trajectories. Using variable, irregular grids is very helpful here, as it provides a variety of length edges. In our cloth training sets, we also randomly rotate the initial cloth mesh, to make sure edges in many orientations are seen in training. One interesting avenue for future research could be to predict uncertainty using e.g. an ensemble model, which could be used to ensure that enough data diversity was present for a certain generalization experiment. Another prominent area for improvement for future papers, as we state in the conclusions, are around adaptivity: being able to optimize the discretization based on a downstream task could improve the performance of the model beyond the ground truth data.
>
> - By nearest neighbor baseline, do you refer to performing message passing between nodes which are spatially close, ignoring the mesh structure? If so, we have tested this (see “Comparison to mesh-free GNS model” in the results section); we find that both mesh-space information and passing messages in the mesh are vital for simulating elastic materials.
>
> - Error for a given accuracy: This is an excellent idea. For the systems we study this might be challenging, as big changes in resolution will alter the system dynamics (e.g. due to numerical viscosity in fluids, and triangle locking in cloth), and make it hard to perform 1:1 ground truth comparisons. However, this is definitely something we want to think about and explore in the future; there might also be interesting tradeoffs for performance/accuracy in the network model itself.

---

### Official Review · AnonReviewer2 · 2020-10-28
**Promising results but lack some ablations and quantitative evaluations**

**Rating:** 6
**Confidence:** 4

**Review:**

Contributions

The authors present a graph convolution architecture suitable for learning physical simulation on meshes.
This method supports dynamically re-meshing to locally adapt to the complexity of the simulated physics, and is stable to relatively long rollouts of physical simulations.

Strong points

The architecture presented is generic and seems suitable for a wide variety of physical tasks. This point is supported by multiple experiments on diverse tasks.
Getting a second architecture to learn the sizing field is very effective. The authors demonstrate a way to supervise this second network even when the sizing filed is not exposed by the ground truth physical simulator. This way is experimentally and quantitatively demonstrated to be robust. As a consequence, the whole architecture remains generic enough to be applicable to emulation tasks even when the simulator is not providing an explicit sizing tensor.

The proposed architecture is surprisingly stable in long rollouts, and does not diverge nor accumulates errors from previous steps. As noted in the Appendix, robustness is enforced via a noisy training scheme from previous literature. Thanks to its convolutional architecture and positional encoding, it can be trained on small systems that are inexpensive to simulate and then later be applied to larger, more complex and fine grained problems. This has the potential to enable fast physical emulation without huge training costs.
Adding one-hot encoding allows to model complex interactions between physical systems, such as an actuator using on an elastic plate (DeformingPlate experiment).

Finally, comparison against existing methods (based on CNN, graph convolutions or particle-based representation) seems favorable to the presented technique.

The paper is well written. Linking to anonymised videos showing experiment results is well done and provides an enjoyable reading experience.

Weak points

Authors do not provide ablations studies on the main novelties of their graph convolution architecture. A fully grounded paper would show the quantitative benefits of:
Adding world edges in addition to existing mesh edges
Adding labels on edges that are initialised with distances
Dynamically remeshing at each step: what happens if you just do not re-mesh? Fig. 3b provides an intuition for it, but I cannot find any quantitative study of it. More than that: in table 1, FlagSimple (fixed meshing) metrics are comparable or better than FlagDynamic. This seems to suggest re-meshing is not useful in this case.
The above 3 points are really what sets this paper apart from the rest of the literature on graph convolution nets. Seeing how they impact results seems mandatory to assess how MeshGraphNet is more than just an existing mesh convolutional network architecture plugged into a physical problem.

About benchmarks: it is not clear why the concurrent method GCN was not taken in full. Its super-resolution part was removed for the purpose of this submission, but it seems to be inherently part of it. Removing this component is a-priori an alteration of its architecture, and the reasons for discarding it are not justified.
Moreover, in comparing to GCN the authors conclude that the key reason for MeshGraphNet to outperform GCN is the positional encoding added to edges, but this claim is not supported experimentally. This could either be done by removing it from MeshGraphNet, or somehow add something similar to GCN.

Since GCN seems good at predicting static values but very quickly diverge for longer rollouts, the two tasks seem quite different. As noted in the Appendix, training noise was added to make the presented method robust to longer rollouts. A fair comparison with GCN would also include this.

Generalization is mostly demonstrated qualitatively : the difference between the training/testing scenarios are not stressed enough - apart for the impressive extrapolation case from Airfoil. For example, the generalisation on FishFlag remains qualitative, but is not quantitatively assessed, and similarly for the Windsock. “Visual similarity” is not 100% satisfying.

Recommandation

Weak accept: I recommend to accept the paper for the promising results it displays, but would like to see the missing quantitative results I mention above addressed - ablation study and experiments.

Questions

In grossly in decreasing order of relevance (first ones = useful to assess the paper / last ones = personal curiosity):
1 - Why was the super-resolution module of GCN removed?
2 - Varying wind speeds/directions on the flag experiment: since each time step is computed separately, to what extent does it really demonstrate generalisation to vary wind direction at each tilmestep, knowing that they were each independently seen during training?
3 - On the Windsock experiment demonstrating generalisation: how much is due to the architecture (convolutions that can by nature be applied to bigger graphs) and how much the network has “learned” the physics?
4 - In Table 1, what is causing t_full to be strictly larger than t_model for the FlagSimple experiment, where no graph re-meshing or re-computation is happening?
5 - What is the chosen time step for 1 prediction? Since it is fixed, how was it chosen?
6 - How are mesh space coordinates u_i’s attributed? Does it matter ?
7 - Does jointly predicting related physical values help? This is something usually observed in the literature, but did you observe similar results? For example in the CylinderFlow experiment you have one network predicting the change in momentum and the pressure field. Does it degrade RMSE to get 2 networks, one for each of these quantities?

Additional recommandations

Clarity could be improved. I make the following remarks, that did not influence on my rating of the paper:
Section 3.1 “Encoder” : I would split the encoding part into two, to clearly mark the difference between initial values provided to the network : (A) one is defined by physical quantities that you provide as input to the mesh vertices (you denote it q_i). The other (B) is more of an internal, method-specific pre-processing step and consists in creating world edges, and labelling world and mesh edges with their length. Presenting (A) and (B) in the same paragraph can be confusing, since they really are not on the same level of processing. (A) is the real, physical input, (B) is already a pre-processing step.
Section 3.1 “Processor” : why denote GraphNet blocks as P_i if it not mentioned anywhere else? Moreover, using _i as an index is confusing since it is elsewhere reserved to graph node indices.
History size h is mentioned in the introduction of 3.1, in 4 (FlagDynamic and SphereDynamic) and in Fig. 5d, but does not fit the formalism presented in 3.1. It should  be presented in a way that accommodates for the possibility of feeding more than just the previous step’s information.
Section 4: could be improved with regards to what is provided as input to the network. For example, u_i is always provided as input, so why repeat it? As far as I understand it, what matters is what you input as physical properties q_i, and what/how you label n_i. In this section q_i are well defined, but for some experiments I have trouble seeing what you encode into n_i - it is only explicated for DeformingPlate.
Section 4: all experiments and datasets are described except for FlagSimple. Is this an oversight?
Globally, clarifying the formalisation (3.1) and values (4) of the network’s inputs would help.

Typos or minor improvements:
3.2, top of page 5: “iff” > “if and only if”
End of A.1 page 13: “… is appended TOO the node” > “to the node”

---

> ### Author Response · Authors · 2020-11-14
> **Response to R2**
>
> Thank you for your comments, questions and suggestions to improve the manuscript.
>
> **Ablations**
> We agree that adding these ablations will add value to the paper, and we have started the ablations mentioned below. We will add their results to the paper when complete.
> - We’re currently running an ablation experiment with/without world edges. Note that on the experiment SphereDynamic it is easy to see why world edges are strictly necessary: the sphere mesh and the cloth mesh are not connected, so without world edges, information on the sphere collisions cannot reach the cloth nodes, and the cloth cannot interact with the colliding sphere.
> - We’re also running an ablation on our model with relative distance encoding in edges, versus encoding absolute quantities in nodes. Without relative encoding, we expect overall lower performance, and significantly weaker generalization to larger domains.
> - Dynamic remeshing at every step is optional, and our method works without it (as shown by the examples FlagSimple, DeformingPlate, Airfoil, CylinderFlow) but doing so can significantly improve the simulation quality. Aside from fig. 3b, you can directly observe this on the video site when looking at ‘Dynamics Remeshing’ and ‘Comparison: GNS’ below. FlagDynamic and FlagSimple share the same simulation setup, but FlagSimple does not use dynamic remeshing, and bends and folds are under-resolved and show jerky motion. We will emphasize this in the text. The RMSE in table 1 measures the error compared to the corresponding ground-truth simulation (statically meshed for FlagSimple, dynamically meshed for FlagDynamic). Since both errors are compatible (and even slightly lower for FlagDynamic), this means our model will see the same, or even slightly more, benefits from adaptive remeshing as the classical solvers, which can be substantial in the case of cloth.
>
> **GCN comparison**
> We realize we did not make this clear enough in the paper, but the intention was to compare to a graph neural network architecture (GCN, which does not compute messages nor uses edge features) different than the one used on GraphMeshNet, but not to compare to the specific full method from [Belbute-Peres et al.]. While there are many papers using GCN-based architectures, we chose the network architecture from [Belbute-Peres et al.] since it shows results on a domain very similar to our Airfoil dataset. We will state this more clearly in the revised version of our paper.
> Our aim in this work was to provide a general-purpose framework which can learn the dynamics of a wide range of systems from data from a black-box solver. The full method in [Belbute-Peres et al.] is a very interesting approach; however, super-resolution has very different goals and properties to our black-box approach, and is not 1:1 comparable. Practically speaking, such a comparison would also be very hard to execute. First, the full method requires a solver in the loop, with access to gradients. This is not readily available with most solvers, e.g. our data sources COMSOL and Arcsim do not provide gradients and aren't easy to run as part of an optimization loop. Second, our method is built for dynamical simulation, while [Belbute-Peres et al.] directly predicts a steady state output. Even for our Airfoil example, which uses the same solver, running a dynamic simulation would require large changes to [Belbute et al.]. E.g., what would ensure continuity in the up-scaled velocity fields? Should the output of every simulation step be fed back as input to the solver, and is this even possible in SU2?
>
> We did however put significant effort to provide a fair comparison to the GCN architecture. We used the same training procedure as in our method, including training noise, ran exhaustive hyperparameter scans and reported the best results obtained. In the result section we also report results on a variant which uses the same node function (MLP with LayerNorm) and number of layers as in our method. Excluding these factors is part of the reason that makes us believe relative encoding in edges is a key reason for the performance difference. As mentioned above, we are also running an ablation on this to clarify the finding.

---

> > ### Author Response · Authors · 2020-11-14
> > **(continued)**
> >
> > **Generalization**
> > We are currently running an additional generalization scan on larger versions of the flag mesh the model was trained on, and will report error numbers for this, as well as the windsock and fish flag experiments in the revised version of the paper. This should allow us to separately measure the effect on the error of scaling up, versus generalizing to a novel shape and to provide insights on the error trade-offs when moving from smaller to larger meshes.
> > For the varying wind speed example, wind speed and direction were kept fixed per trajectory in training. The dynamics under changing wind speed/direction, e.g. the falling flag at the end of the video, are qualitatively different from constant, low wind speed. This intends to demonstrate that for learning the full dynamics, it’s sufficient to provide some fixed samples of a global parameter, instead of exposing the model to all possible combinations of transitions (low-to-high speed, wind left-to-right etc.) in the training set.
> > Finally, on the question regarding “learning physics”: To exhibit strong generalization as in the windsock example (generalizing from a rectangle domain to complex curved shapes of much larger scale), the model necessarily has to learn at least a subset of the underlying physics. As you mention, part of it is not purely learned from data, but induced by our architecture (which is e.g. by construction translation-invariant due to relative encoding; absolute positional encoding would not share this property), and part of it is learned, but in a way that encourages generality (locality, and training on a variety of triangle sizes). We believe that this combination of very general biases and compositional learned elements are key to making learnable simulations more practically useful.
> >
> > **Further Questions and comments**
> > - Yes, you are right, t_full and t_step should be the same for these experiments. The reason for the difference is likely they were measured slightly differently: t_full was measured in rollout, i.e. the model output being fed back as input for the next step, while t_step is measured by directly feeding the steps from ground truth tractories into the model, which might result in slightly different GPU execution times, in the order of ~1ms. We will therefore round the numbers to 1ms precision, to avoid the impression of higher measurement accuracy.
> > - We will add time step information to the manuscript. Time steps were chosen such that we could cover enough interesting dynamic behavior in a few hundred time steps (to keep dataset size and training time manageable); we didn’t explore different time step sizes beyond this.
> > - The mesh space coordinates u_i are sometimes provided by the solver. If not, any relaxed state of the mesh will work: often the first frame of simulation is in relaxed configuration, and in that case the world coordinates of frame 1 can be used as mesh coordinates. Due to our model’s translation invariance, absolute positioning in mesh-space does not matter.
> > - We didn’t observe a performance benefit from predicting auxiliary fields. E.g. on CylinderFlow, the velocity error when predicting only velocity, vs. velocity+pressure, was very similar.
> > - Thank you for your detailed suggestions on improving paper clarity. We will do a pass over section 3 and 4, and incorporate your input.

---

### Official Review · AnonReviewer4 · 2020-10-28

**Rating:** 6
**Confidence:** 4

**Review:**

Summary:

The paper presents a scheme for applying graph networks to surface and volume meshes for simulation of fluid flow and structural analysis.
To this end, the method adds problem specific positional features to the mesh nodes and uses a message passing approach to compute the desired outputs.
For Lagrangian systems the discretization is augmented by adding additional edges between nodes that fall below a distance threshold.
The experiments demonstrate better performance than grid-based or other graph-based approaches on the selected problems.


Score:

I am on the fence here. I think the paper has some good results but only scratches some important questions with respect to the mesh.
Further, claims like computational efficiency are only weakly supported.
I hope my questions and concerns in can be addressed.


Pros:

* The videos are good and show the stability of the method and the generalization extent.

* The network architecture is straightforward and simple.
  Learning the sizing field is a good addition and a good future direction for adapting the mesh.

* The experiments show well the advantages of adaptive grids versus regular grids and meshless methods.


Cons:

* The claim that this approach can be used to speed up simulations is not well supported.
  Table 1 gives the inference time per frame but does not give any information about the hardware used in the experiments.
  (The appendix mentions a V100 GPU but I could not find information about the CPU which is used for the GT simulation.)
  The comparison would also be more interesting if GPU solvers would be added to the comparison.
  As an alternative the graph network can be run on the CPU.

* The appendix contains some information about the remesher for the cloth simulation but apart from that meshing is not studied in the paper despite being crucial for this method.
  The meshes in the datasets use triangles and tetrahedra but another possible choice can be quads and hexahedra.
  I would be valuable to know if the proposed method can generalize to this, needs to be trained for this specifically or does not work at all in this case.
  A similar direction would be to experiment with badly shaped elements that are large but have small area/volume to check if the learning approach is more robust.

* The related work should be extended with more works that share similar goals (e.g. improving runtime).
  - Capuano and Rimoli, "Smart finite elements: A novel machine learning application" Computer Methods in Applied Mechanics and Engineering 2019.
  - Luo et al., "NNWarp: Neural Network-Based Nonlinear Deformation" IEEE Transactions on Visualization and Computer Graphics 2018.


Questions:

The description of the world-space edges is not clear to me. Since r_w is an absolute threshold, can this lead to crossing edges in extreme situations?

The "Computational efficiency" section mentions that the network can do much larger time steps.
What is the time step used during inference for the network? What is the time step for the classical simulator?

Can you give a bit more information about the elements used in the DeformingPlate dataset? Are these linear elements?

The flag videos show generalization with respect to rotation about the gravity axis.
Is the network solution invariant with respect to all 6 DOF?

What is the memory footprint of the method? Can problems with millions of nodes be simulated on a recent GPU?

---

> ### Author Response · Authors · 2020-11-14
> **Response to R4**
>
> Thank you for your comments, questions and suggestions to improve the manuscript.
>
> **Performance**
> We acknowledge that the claims about timing and speedups need further clarification, and we will include a more detailed report in the paper.
> The ground truth solvers (Arcsim, COMSOL, SU2) are popular, highly optimized solvers, which use CPU multiprocessing, but do not support GPU acceleration. We ran them on a 8-core workstation CPU. It’s quite common for general purpose simulation packages in this domain to choose CPU, not GPU solvers. GPUs can offer large speedups for specialized setups (say, solving elliptic PDEs on uniform grids with simple boundary conditions), but the benefits of GPUs are much reduced for simulations on complex irregular meshes with non-trivial boundary conditions, as components like tree-based spatial lookups, AMG, coupled solvers, etc. are hard to port to GPUs, and yield lesser speedups.
> One of the advantages of our method is precisely that by using standard NN building blocks, we can exploit the benefits of hardware accelerators in domains where this is hard for classical solvers. For a user with a GPU, this means our approach can in practice be much faster than classical solver for these problems. And while we train and evaluate on a single GPU, model parallelism enables scaling up to even larger domains and higher performance on e.g. a highly interconnected TPU pod, with very little code changes.
>
> **Meshing**
> These are good questions which may be of interest to other readers as well, so we will clarify our position in the paper.
> We demonstrated and studied adaptive remeshing on FlagDynamic and SphereDynamic, since adapting resolution at each step is particularly beneficial for this domain, and much less so for the other examples studied. However, there is good reason to believe that our approach would work for other adaptively remeshed simulations. A.3.1, 3.1 allow us to estimate targets and learn a sizing model simply based on the observed edge length distribution, which is not specific to our examples. The non-learned local remesher will of course be different depending on the type of element used; for tetrahedra, e.g. an approach as in [Wicke et al., Dynamic local remeshing for elastoplastic simulation] could be used. The same may be true for simulation data with quads or hexahedral elements, although an easier solution would be to simply divide each quad into two triangles (or hex to tetrahedra), and then directly use our method as described in the paper. More broadly, remeshing is an exciting direction worth studying further, and we’re very interested in exploring e.g. how to perform full remeshing without a non-learned component, or how to optimize the meshing for a downstream task, but this is a bit outside the scope of this paper.
>
> **Other comments and questions**
> - Thank you for the additional references, we will incorporate them into the paper
> - We will clarify the role of world-space edges in the paper. We incentivize the network to use mesh-edges for computing internal dynamics (which may involve integrating over mesh elements), and world edges to compute external dynamics, by the way we structure connectivity, and by storing different information on them (e.g. world edges don’t have access to mesh coordinates). Computing external dynamics (e.g. collision) does not require the edges to form non-overlapping elements, so crossed world edges should not be a problem.
> - We will add the timestep size used in our model to the paper. While the output timestep of the simulators matches ours, the number of internal time steps taken by the ground truth simulators vary strongly due to e.g. adaptive stepping; they are in the order of 10-1000 steps per one step of our model.
> - Yes, DeformingPlate uses linear elements, we will add this information to the manuscript.
> - Our model is translation-invariant by construction. But due to the high amount of diversity in the individual triangles’ orientation, it typically also generalizes well to orientation changes. In the rotating flag example, we trained on different wind speeds and directions, but these parameters were kept fixed per trajectory in training. At test time, the model was able to generalize to arbitrary transitions between these values, without seeing any transitions in training. E.g. The falling flag at the end of the video due to stopping wind is qualitatively very different from anything seen in training.
> - Memory and performance of the core model is linear in the number of edges/nodes. Fitting millions of nodes on a single GPU may be hard, but since we use generic NN building blocks, model parallelism would allow us to distribute memory and compute over several accelerators, to scale to a higher number of nodes.

---

> > ### Comment · AnonReviewer4 · 2020-11-16
> > **Minor comments and hints for the revised version**
> >
> > Thank you for your response.
> >
> > **Performance** The motivation is good. I am looking forward to the additional information. Please be aware in your argumentation that GPU solvers do exist (ANSYS) and tasks that have previously been difficult for GPUs like spatial lookups in raytracing now even have special hardware support. Please add inference times for the 8-core CPU and information about the memory used for the simulations.
> >
> > **Meshing** The title of the paper is very general but there are many different types of meshes. Therefore, it is important to state that the models/experiments are limited to well shaped tetrahedral meshes for simplicity. Exploring the meshing further is definitely a good direction for future work.
> >
> > **Rotational invariance** As I understand the model is not rotationally invariant but generalizes well in the flag experiment due to the high amount of diversity at a local scale.

---

### Official Review · AnonReviewer1 · 2020-10-29

**Rating:** 9
**Confidence:** 4

**Review:**

- Summary

This paper presents a graph-network-based architecture for learning to perform mesh-based simulations, which can be run more efficiently than the full, "ground-truth" simulations. The experiments demonstrate that the proposed method is able to learn to simulate a wide range of different physical scenarios. Moreover, the presented results also demonstrate an ability to generalize to configurations different from the one seen in training.


- Pros

The paper is clearly written and strikes a good balance of presenting the relevant information in its main body and including the necessary details in the supplementary material. The additional videos provided are helpful in demonstrating the model's capabilities.

The experiments performed are comprehensive, with comparisons to a wide variety of significant baselines (e.g., other graph neural networks, grid based methods, etc.) and evaluations of the impact of diverse hyperparameters.

Most importantly, the results presented are strong, containing both impressive qualitative demonstrations and good quantitative performance when compared to baselines.

The ability to operate directly on meshes, in contrast to previous work that operated on structured grids, makes this method more relevant to real-world applications.


- Cons

The proposed method is, conceptually, a straightforward evolution of previously proposed graph-network-based methods for physics simulations, with most notable similarities to Sanchez-Gonzalez et al. (2020) ([33] in-paper reference).

The ability to generalize to higher-dimensional meshes is only briefly (quantitatively) evaluated (more on this in the comments below).


- Reasons for score

Overall, given the "pros" described above, notably the strength of the results presented on a wide variety of tasks, I recomend this paper for acceptance.
As mentioned above, the method proposed is a modification of a graph-network based method previous employed to simulate particle dynamics. In order to work on meshes, some modifications are proposed. such as adding mesh- and world-space connections to the graph, learning to predict the S matrix to remesh, etc. As such, this works stand on the strength of the practical results it achieves. The results demonstrate an impressive ability of the proposed method to learn to capture both Eulerian and Lagrangian mesh simulations. The wide diversity of tasks for which the model works well is unprecedented. The experimentation is extensive, with comparisons to significant baselines and evaluations of the relevance of different hyperparameter choices.

Moreover, methods that learn to simulate physical processes more efficiently, and applications of graph neural networks are two research directions that have garnered a lot of interest in recent work. As an intersection of these two areas this paper should be of interest to a wide audience in the conference.


- Additional comments

One important drawback of these types of physical simulation methods, which require training on ground truth data, is that training itself requires a large amount of data from the very process we want to simulate. If the learned model only presents a limited ability to generalize, this can severely limit the applicability of the method. This does not seem to be the case here, as the experiments seem to demonstrate an good ability to generalize to unseed scenarios, provided in part by the ability of the graph neural network to learn local/scale-invariant interactions.
In this direction, an important capability of the model, suggested in the paper in the "tassel sub-experiment" of the FlagDynamic experiment, is the ability to generalize to much larger meshes than the ones seen in training. As mentioned in the "cons" section above, however, this ability is only briefly (qualitatively) evaluated. It would be interesting to see a more thorough evaluation of this ability, due to its potential large importance to practical applicability, by allowing the model to be trained on smaller examples and then scaled up at inference time.
Moreover, a thorough analysis of the error trade-offs when moving from smaller to larger meshes would also be interesting.

In the section on "Key hyperparameters", within the experimental results, it is claimed that "the model performs best given the shortest possible history ..., with any extra history leading to overfitting. This differs from GNS [33], which used $h \in 2 \dots 5$ for best performance." Do the authors have any hypothesis as to why this might be the case here, versus the observation in [33]?

---

> ### Author Response · Authors · 2020-11-14
> **Response to R1**
>
> Thank you for your comments and suggestions on improving the manuscript.
> - We agree that quantitative results for running on larger meshes at inference time would strengthen the results. We are running an additional generalization scan on larger versions of the flag mesh the model was trained on, and will report error numbers for this, together with the error for the windsock and fish flag experiments. This should allow us to separately measure the effect on the error of scaling up, versus generalizing to a novel shape and to provide insights on the error trade-offs when moving from smaller to larger meshes, as suggested. We will include these additional experiments.
> - Besides its ability to generalize well, another advantage is that our model is comparably data-sparse; our examples were trained on only 1000 trajectories.
> - On your question regarding performance with respect to GNS’ history length, we believe that this may be due to availability of information about the material’s rest state, which is necessary to compute elasticity and plasticity. Mesh-space coordinates and connectivity allow our model to explicitly compute these quantities; for GNS however, the relaxed material state has to be inferred from history (think e.g. finding the rest length of a spring by observing a few oscillations). Hence, GNS needs longer history input for elastoplastic materials, while our method only requires the minimal set. This minimal history improves stability & generalization in our model, as providing the model with history it does not need makes the model slightly more prone to overfitting.

---

> > ### Comment · AnonReviewer1 · 2020-11-24
> > **Response**
> >
> > Thanks for the response. The answers presented here clarify my previous questions, and the additions to the paper are very welcome. I will maintain my (already positive) evaluation of the paper.

---

### Author Response · Authors · 2020-11-20
**Revised version**

We have revised our paper and supplemental, and uploaded a new version to OpenReview. We have run several new experiments, and added references, edits and clarifications suggested by the reviewers. All new or changed text is marked pink in the document.

Specifically, we have:
- run two additional ablations: We ran an ablation of our model without world edges, and a version with absolute, instead of relative encoding. Both versions perform significantly worse than our main model. We have added these results to the paper.
- reported quantitative error numbers for our generalization experiments. In addition, we also ran models on several scaled-up versions of FlagDynamic and WindSock, reported errors, and discussed these new results.
- timed our model on the workstation CPU the ground truth solvers were run on. Even in the CPU-only setting, our model is 4-22x faster than the ground truth solvers. We have also added a more detailed report on performance to the manuscript.
- clarified the language around the GCN comparison to avoid false impressions
- added information on time stepping
- run a model for an extended amount of time (40000 steps, 100x longer than the training trajectory), and added a video to the site. The rollout remains stable for the entire duration.
- added references and several other clarifications raised in review

We believe this revision should address all points raised in the review.

---

### Decision · Program_Chairs · 2021-01-07
**Final Decision**

**Decision:**

Accept (Spotlight)

**Comment:**

The paper develops a methodology for using graph neural networks for mesh-based physics simulation. This extends recent work that focused on grids or particles to mesh-based domains, which are challenging due to irregular (and possibly changing) connectivity. The reviewers had some concerns but recognized that this is an important work that will be of broad interest and may have significant impact.